

# Testing the effects of perimeter fencing and elephant exclosures on lion predation patterns in a Kenyan wildlife conservancy

Marc Dupuis-Desormeaux[1], Zeke Davidson[2], Laura Pratt[2], Mary Mwololo[3] and Suzanne E. MacDonald[4]

[1] Department of Biology, York University, Toronto, Ontario, Canada
[2] Conservation Department, Marwell Wildlife, Winchester, Hampshire, United Kingdom
[3] Research Depeartment, Lewa Wildlife Conservancy, Isiolo, Kenya
[4] Department of Psychology, York University, Toronto, Ontario, Canada

Corresponding author
Marc Dupuis-Desormeaux,
marcd2@me.com

## ABSTRACT

The use of fences to segregate wildlife can change predator and prey behaviour. Predators can learn to incorporate fencing into their hunting strategies and prey can learn to avoid foraging near fences. A twelve-strand electric predator-proof fence surrounds our study site. There are also porous one-strand electric fences used to create exclosures where elephant (and giraffe) cannot enter in order to protect blocs of browse vegetation for two critically endangered species, the black rhinoceros (*Diceros bicornis*) and the Grevy's zebra (*Equus grevyi*). The denser vegetation in these exclosures attracts both browsing prey and ambush predators. In this study we examined if lion predation patterns differed near the perimeter fencing and inside the elephant exclosures by mapping the location of kills. We used a spatial analysis to compare the predation patterns near the perimeter fencing and inside the exclosures to predation in the rest of the conservancy. Predation was not over-represented near the perimeter fence but the pattern of predation near the fence suggests that fences may be a contributing factor to predation success. Overall, we found that predation was over-represented inside and within 50 m of the exclosures. However, by examining individual exclosures in greater detail using a hot spot analysis, we found that only a few exclosures contained lion predation hot spots. Although some exclosures provide good hunting grounds for lions, we concluded that exclosures did not necessarily create prey-traps per se and that managers could continue to use this type of exclusionary fencing to protect stands of dense vegetation.

## INTRODUCTION

Conservancies that host endangered species, such as black rhinoceros (*Disceros bicornis*) and Grevy's zebra (*Equus grevyii*), go to great lengths to provide adequate habitat, food resources and security for their wildlife to ensure long-term viability. These conservancies

protect rhinoceros by erecting multi-strand electrical perimeter fences as well as using a multitude of other security measures, including armed anti-poaching patrols and aerial surveillance. These perimeter fences are also effective at segregating wildlife from the surrounding agricultural and pastoral lands. In many parts of Africa, including Kenya, there is an increased reliance on fencing to protect wildlife and reduce human-wildlife conflicts (*Kioko et al., 2008*; *O'Connell-Rodwell et al., 2000*; *Sitienei, Jiwen & Ngene, 2014*; *Thouless & Sakwa, 1995*).

Fencing wildlife can cause direct mortality if animals become entangled and killed while attempting to cross the fence (*Albertson, 1998*; *Harrington & Conover, 2006*; *Mbaiwa & Mbaiwa, 2006*) or can cause indirect mortality, when predators, such as wild dog (*Lycaon pictus*), hunt near fences, as escape routes are limited (*Davies-Mostert, Mills & Macdonald, 2013*; *Romañach & Lindsey, 2008*). Further, fenced reserves are essentially closed systems, which can provide advantages to predators that can quickly deplete prey populations. For example, *Tambling & Du Toit (2005)* found that the ratio of lion to prey population could be three times higher inside a fenced reserve.

Fencing also has secondary drawbacks that can affect long-term herbivore population viability, including reduced access to resources (*Brenneman et al., 2009*; *Loarie, Aarde & Pimm, 2009*; *Olsson & Widen, 2008*), and the creation of edge effects (*Massey, King & Foufopoulos, 2014*; *Newmark, 2008*; *Vanak, Thaker & Slotow, 2010*). *Vanak, Thaker & Slotow (2010)* found that perimeter fences in a completely fenced reserve affected elephant foraging movement up to 3.8 km inside the fencing irrespective of habitat composition. Effective protection of species is usually highest deeper inside a protected reserve, where human encroachment is less likely. Human wildlife conflict near reserve edges are the principal cause of mortality and border areas can become population sinks and therefore reduce the density of wildlife populations near the edges (*Woodroffe & Ginsberg, 1998*).

Our study site displays characteristics of both an open and a closed system as it has a semi-porous perimeter fence where three migratory gaps (20–30 m wide) have been created in the perimeter fence to allow wildlife to move freely between the protected area and neighboring multi-use landscape (*Dupuis-Desormeaux et al., 2015*). The first objective of this study was to investigate the effect of perimeter fencing on predation patterns in a semi-porous reserve, testing whether predators have learned to take advantage of perimeter fencing to increase hunting success. Anecdotal evidence of preyed upon carcasses near the perimeter fence suggested that predators might be finding success hunting at the boundary of the conservancy. We tested the hypothesis that prey were killed in proportionally higher numbers near the perimeter fence than elsewhere in the conservancy.

We also investigated the role of elephant exclosures in predation. The presence of mega herbivores, such as elephant (*Loxodonta africana*) and giraffe (*Giraffa camelopardalis reticulata*) can substantially change the availability of browse vegetation accessible to browsers. The elephant's ability to change woody areas into grasslands by felling trees, breaking branches, and debarking trunks is well documented (*Hatton & Smart, 1984*; *Haynes, 2011*; *Mosepele et al., 2009*; *Naiman, 1988*; *Nasseri, McBrayer & Schulte, 2011*; *Pringle, 2008*; *Valeix et al., 2011*). In arid savannas, and in fenced habitat, elephant browsing
can have severe negative effects on woody vegetation (*Guldemond & Van Aarde, 2008*). Giraffe are heavy browsers of tree canopies that result in the suppression of growth, and in combination with elephant and black rhino can have long-term detrimental effects on woody vegetation regeneration (*Birkett, 2002*; *Bond & Loffell, 2001*; *Smart, Hatton & Spence, 1985*). This risk of depleting woody resources to the detriment of the protected browsers can cause management to take proactive measures by creating exclosures, areas that allow free movement of some species while keeping others out. The use of elephant exclosures by managers seeking to protect critical vegetation has been effective and has become more widespread throughout the continent (*Lagendijk et al., 2011*; *Lombard et al., 2001*; *Slotow, 2012*).

However, the success of exclosures at protecting vegetation can attract both browsers and grazers as vegetation outside the exclosures can become more depleted by the foraging habits of the excluded mega herbivore population. Lions prefer hunting where prey is easier to catch rather than where prey is more abundant (*Hopcraft, Sinclair & Packer, 2005*) and so lion intensify their prey search in bushed grasslands and near water (*Davidson et al., 2012*; *Valeix et al., 2010*). At our study site, previous work by *Pratt (2014)* found that the location of Grevy's zebra carcasses killed by lions were clustered in denser vegetation and near watering holes. Given the enhanced nutritional characteristics of the vegetation, it is probable that prey might be preferentially attracted to forage within these exclosures, which may then act as prey-traps. Thus, the second objective of this study was to test if predation events were over-represented either near or inside the exclosures, i.e, do elephant exclosures become prey-traps. This question is important to managers invested in the care of critically endangered species (such as rhinoceros and Grevy's zebras) that are attracted by the browse quality inside exclosures. Thus determining if exclosures are functioning as prey-traps for these species is crucial.

## METHODS

### Study site

We conducted our study at the Lewa Wildlife Conservancy (Lewa) in Isiolo, Kenya (0.20°N, 37.42°E). The habitat at Lewa consisted of Northern Acacia-Commiphora Bushlands and Thickets with an Afromontane section (*White, 1983*)  with significant areas of savannah. Lewa was initially a cattle ranch (1920–1983) and in 1983, in response to declining black rhinoceros population, management converted 2,000 ha to a rhino sanctuary. This sanctuary grew over the subsequent years and in 1995, Lewa officially converted all of its 25,000 ha and upgraded its perimeter fence to a 142 km long, two-meter high electric fence, consisting of twelve-strands of alternating live electrical and grounded wires. The primary purpose of the fence was to segregate the wildlife from the neighbouring communities, thereby reducing human wildlife conflicts. The perimeter fence was continuous except for a few manned gates for vehicle traffic and three purpose-built wildlife gaps created to permit the safe movement of migratory species in and out of the conservancy. There was one 30 m-wide fence gap to the north leading to a pastoralist community, one 20 m-wide fence gap to the west leading to a neighbouring reserve and one 20 m-wide fence gap to the

 

south (20 m wide) leading to a 14 km long elephant corridor connecting to Mount Kenya. The perimeter fence was patrolled daily and maintained by teams of rangers and workmen.

There were 23 exclosures at the study site consisting of areas where single or double-strand electrical wires were set at a height of approximately 1.7–2 m, permitting rhino, zebra and other wildlife to pass underneath, but excluding elephant and giraffe. Both the perimeter fence and the exclusionary fencing had voltage maintained at between 5.0 and 9.0 kV, levels suitable to keep large animals away without causing permanent injury. The locations of the exclosures were chosen based on a combination of factors, including to protect the remaining stands of woody vegetation from elephant (and giraffe) damage, to promote the recovery of woody vegetation, to maintain the aesthetic value of some areas and to protect residential areas. Given that exclosures were not randomly placed on the study site but were targeted to areas that already had vegetation to protect, vegetation cover and the presence of exclosures are confounded variables.

These exclosures have been in place between 4 and 22 years (mean = 14.3 ± 4.4 years) and cover approximately 12% of the conservancy. The creation of these exclosures has allowed a different and a denser woody vegetation mix to develop (*Baker del Aguila, 2010*; *Giesen, Giesen & Giesen, 2007*) and the recruitment of tree saplings was significantly higher inside exclosures (*Baker del Aguila, 2010*). In exclosures protecting riverine habitat, tree cover has become significantly denser over the years, increasing from under 10% in 1979, to 25% in 2000 and to more than 50% in 2006 (*Giesen, Giesen & Giesen, 2007*). However, the creation of exclosures at our study site has deflected browsing pressure to the unprotected habitat and has led to severe decline of unprotected woody vegetation (*Giesen, Giesen & Giesen, 2007*). To this point, fixed-point photography used at the study site has revealed stark visual changes in vegetation density (see Figs. 1A and 1B).

In 2014, Lewa reported approximately 500 elephants in the conservancy (*Mutinda et al., 2014*) and approximately 225 giraffe (*Giraffa camelopardalis reticulata*). All necessary permits were obtained for the described field study from the appropriate agencies (Kenya Wildlife Service Affiliation, KWS/BRM/5001, and Kenyan National Council for Science and Technology, NCST/RRA112/I/NIASI).

## Mortality data

We collected predation data from Lewa dating back to 2004. The collection of carcass data is a strategic imperative for all 283 field staff at the study site and, as of 2015, included anti-poaching patrollers, rhinoceros rangers, safari guides, fencers, trackers, herders, and research personnel. Anti-poaching patrols ($n = 62$, as of 2015) are armed and can travel by foot or by vehicle. These patrollers are deployed 24-hours a day, seven days a week and patrol mostly near the access points, roads, fence-line and fence-gaps but can access all of the study site, including by vehicle, plane and helicopter. They also patrol the entire fence-line searching for security breaches and elephant damage to the electrical fence on a regular basis. On Lewa, rhinoceros rangers ($n = 81$) track on foot each individual black rhinoceros, 24 h a day and report position on a daily basis via radio. These rangers follow the rhinoceros continuously throughout the conservancy. Safari guides ($n = 32$) work mostly out of vehicles but also on foot and horseback and report kills when they discover

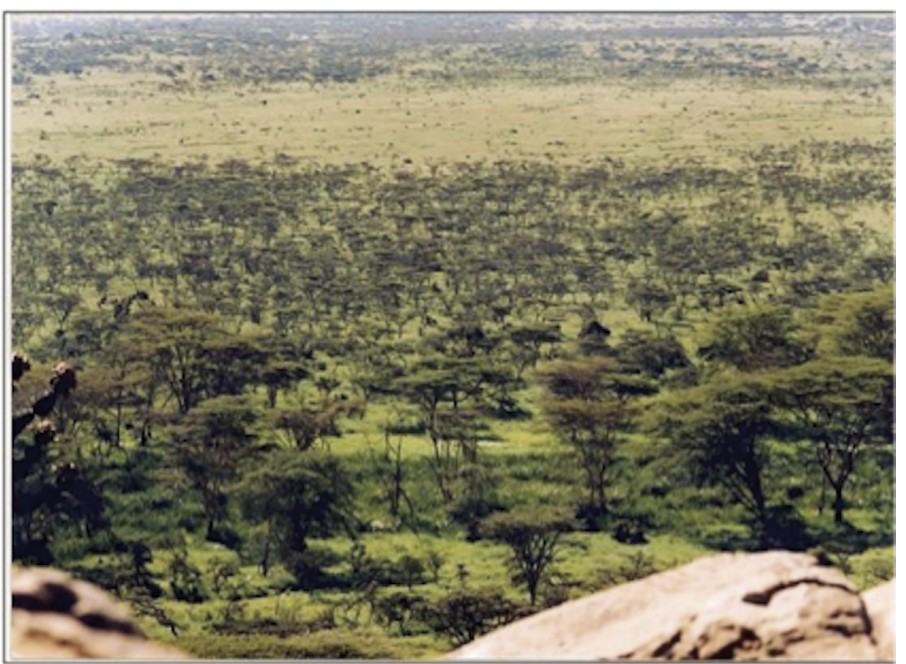

A. View from Craig House 1990 (Lewa)

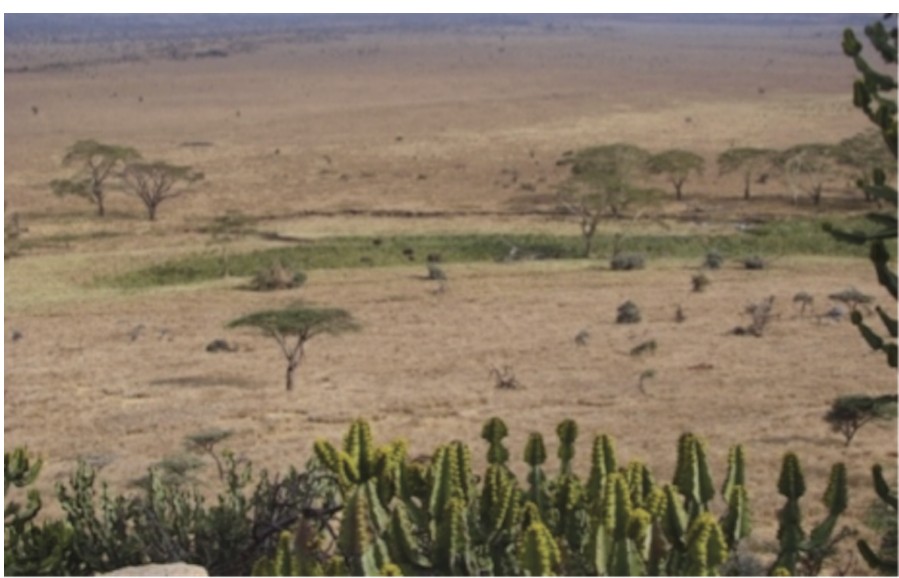

B. View from Craig House 2015 (Lewa)

**Figure 1** **Changes in vegetation cover between 1990 and 2015. Lewa.**

them. It is important to note that the safari guides are highly incentivized to locate fresh kills for visiting tourists. There are also fencers ($n = 38$) that patrol the 150 km perimeter and 98 km exclusion fences for damage. Trackers ($n = 2$) read the spoor of mammals that have used the fence-gaps on a daily basis and report any carcasses found near the fence-gaps. Because the study site supports local cattle herds at various times of the year, herders ($n > 55$) also come across carcasses when tending their cattle. Finally, researchers ($n = 13$) at the study site actively monitor rhinoceros, elephant, ungulates and predators by road vehicle or by plane. It is also noteworthy that the study site has an extensive road network (in excess of 800 km of internal tracks and roads) and that no point is more than 1 km away from a track or road. The study area is thus well monitored and mortality data are considered representative. Despite these efforts, the detection of carcasses in the field is sensitive to prey size and habitat type, given that larger carcasses in more open habitat are easier to detect and last longer, increasing the probability of their discovery. Therefore, smaller prey and prey in very dense habitat was likely under-represented in our sample (*Davidson et al., 2013*).

We established the location of the predation events by assigning a set of GPS coordinates to the descriptive physical locations of each reported carcasses (with a tolerance of $\pm 50$ m). We used only verified predator kills in our data set. We assigned a cause of death to every carcass found and animals that had died of other causes (unverified as predator kills, drought, electrocution, etc.) were not included in our analysis.

Starting in April 2014, Lewa began using an additional technique to locate carcasses of prey killed by lion. The search method is based on the nearest neighbour cluster point method, as described in *Davidson et al. (2013)* where researchers collared five lion groups with GPS radio collars and monitored potential kill sites by identifying locations where the lion activity clustered in excess of four hours. Because of the predominance of lion kills in our data set, we limit the analysis to lion kills (lions, cheetah and leopards representing approximately 81%, 8% and 6%, resp. of the kills in our data).

## Proximity analysis

We performed a proximity analysis to investigate the spatial distribution at the conservancy level. We tested the hypotheses that the level of kills near the perimeter fencing and inside the exclosures were not significantly different than what could be expected elsewhere on the conservancy by comparing the actual density of kills inside the selected zones with the expected density of kills in the rest of the conservancy (using Pearson's chi-squared test).

We created buffers of 100 m and 500 m inside the perimeter fence and intersected those buffers with the shape of the conservancy to calculate the area captured. We then measured the number of kills recorded in each buffer and compared that number to the expected number of kills based on area. With regards to exclosures, we measured both the number of kills inside the exclosures and also the area including a 50 m buffer and compared to the expected number of kills based on the calculated area of the exclosure. *Stander (1992)* reported that the majority of lion kills (73%) were recorded after short chase distances (less than 20 m) and the rest (27%) were between 20 m and 150 m. *Scheel (1993)* observed hunting distances of up to 200 m. We chose to include a buffer distance of 50 m, to allow

for a lion ambush somewhere inside the exclosure but where the actual kill might have terminated just outside the exclosures.

We also calculated a prey selectivity index following *Hayward & Kerley (2005)* using the Jacob's selectivity index (*Jacobs, 1974*) of the prey killed inside the exclosures versus available prey (based on the mean annual census numbers collected from 2004 to 2014 by Lewa staff using standardized aerial and ground surveys), and we compared it to the selectivity index for lion kills outside of the exclosures. Jacobs' selectivity index is calculated as follows:

$$D = \frac{r - p}{(r + p) - 2rp}$$

where $r$ is the proportion of the total kills for a particular species at the study site and $p$ is the proportional availability of this prey species. The index ranges from $-1$ to $+1$, where negative values represent relative avoidance ($-1$ being complete avoidance) and positive values represent relative preference ($+1$ being complete preference).

## Hot spot analysis

In order to investigate the spatial processes at a local level, we performed a hot spot analysis on the aggregated lion predation data using a local statistic (Getis-Ord Gi*). First, we aggregated predation locations that were within a 100 m radius of each other in order to create a set of weighted features necessary for the analysis. We then calculated a global measure of spatial autocorrelation (using a Global Moran's I statistic) between the aggregated predation locations where the Global Moran's I returned a $Z$-score, positive if the spatial distribution was more spatially clustered than would be expected under a random spatial process, negative if more dispersed. We calculated the spatial autocorrelation for a number of incremental neighbourhood sizes, starting with the minimum neighbourhood where every location has at least one neighbour. We selected the neighbourhood size that corresponded to the first peak in significant $Z$-scores (i.e., where the next incrementally larger neighbourhood had a smaller $Z$-score), thus selecting a neighbourhood size that corresponded to the smallest distance where significant spatial autocorrelation was occurring, i.e., the first distance where the spatial processes promoting clustering were more pronounced. We used this neighbourhood size as the distance input factor of the hot spot analysis.

The hot spot analysis is sensitive to which conceptualization of spatial relationship is used when searching for neighbouring points of influence. We used a zone of indifference conceptualization, a hybrid of fixed distance and inverse distance conceptualizations, where the influence of locations inside a fixed neighbourhood distance are weighed equally and where the influence of locations that fall outside that distance are reduced in proportion to the distance away from the neighbourhood boundary. The hot spot analysis returned a local $Z$-score (Getis-Ord Gi* statistic) for each aggregated location where significantly positive scores indicated locations that were clustering with other locations of high predation (hot spots) whereas negative scores indicated clustering of locations with low predation counts (cold spots). We then examined if any of the hot spots fell within the perimeter buffers

**Table 1    List of top ten predated carcasses at the Lewa Wildlife Conservancy, 2005–2014.**

| Common name | Species | Lion kills at study site | Lion kills inside exclosures | Lion kills inside exclosure + buffer (+50 m) | Lion kills inside 100 m of perimeter fence | Lion kills inside 500 m of perimeter fence |
|---|---|---|---|---|---|---|
| Plains zebra | *Equus quagga* | 253 | 43 | 49 | 13 | 42 |
| Grevy's zebra | *Equus grevyi* | 127 | 26 | 28 | 4 | 14 |
| Reticulated Giraffe | *Giraffa camelopardalis reticulata* | 60 | 6 | 9 | 5 | 9 |
| Eland | *Tragelaphus (Taurotragus) oryx* | 43 | 6 | 8 | 3 | 5 |
| Buffalo | *Syncerus caffer* | 29 | 4 | 4 | 0 | 3 |
| Waterbuck | *Kobus ellipsiprynmus defassa* | 28 | 10 | 11 | 1 | 2 |
| Warthog | *Phacochoerus africanus* | 25 | 12 | 13 | 1 | 2 |
| Impala | *Aepyceros melampus* | 23 | 8 | 8 | 0 | 0 |
| Beisa Oryx | *Oryx gazella beisa* | 11 | 2 | 2 | 0 | 1 |
| Hartebeest | *Alcelaphus buselaphus* | 6 | 0 | 0 | 1 | 1 |
| All others | | 23 | 4 | 6 | 1 | 3 |
| Total | | 628 | 121 | 138 | 29 | 82 |

and exclosures. Spatial analysis was done in ArcMap 10.3 (ESRI, Redlands, CA, USA) and statistical analysis was performed in SPSS (IBM Corp., Armonk, NY, USA).

## Vegetation survey

In order to better understand the role of vegetation within the exclosures, we performed a supervised classification of the vegetation cover using imagery gathered from Landsat 8 satellite images with a 30 m resolution. We matched the supervised classification with on the ground plot survey data collected by *Giesen, Giesen & Giesen (2007)* and created five bands of vegetation cover, ranging from grasslands (tree cover less than 2%) to forests (tree cover larger than 40%). We then tabulated the number of kills that fell within each zone and compared the actual kills versus expected kills given the area covered by each zone in order to get a better understanding of lion hunting habitat preferences.

## RESULTS

We recorded 772 kills that could be attributed to a specific predator species over a ten-year period, including 628 identifiable lion kills. The top ten preyed upon species within the study site, and their representation within the perimeter buffers and the exclosures (including a 50 m buffer), are shown in Table 1.

Of those 628 lion kills, 29 were located within 100 m of the perimeter fence, an additional 53 were inside the next 400 m, 121 were located inside exclosures and an additional 27 kills were within a 50 m buffer zone outside the exclosure perimeter. We mapped the individual kills in relation to the perimeter fence and exclosures in Fig. 2.

Predation rates near the perimeter fencing as a whole were in proportion or lower than elsewhere in the whole of the conservancy. The number of kills inside the 100 m perimeter fence was not significantly different than a random distribution given the area represented. However, within the 500 m buffer zone, the number of kills was significantly lower than expected.
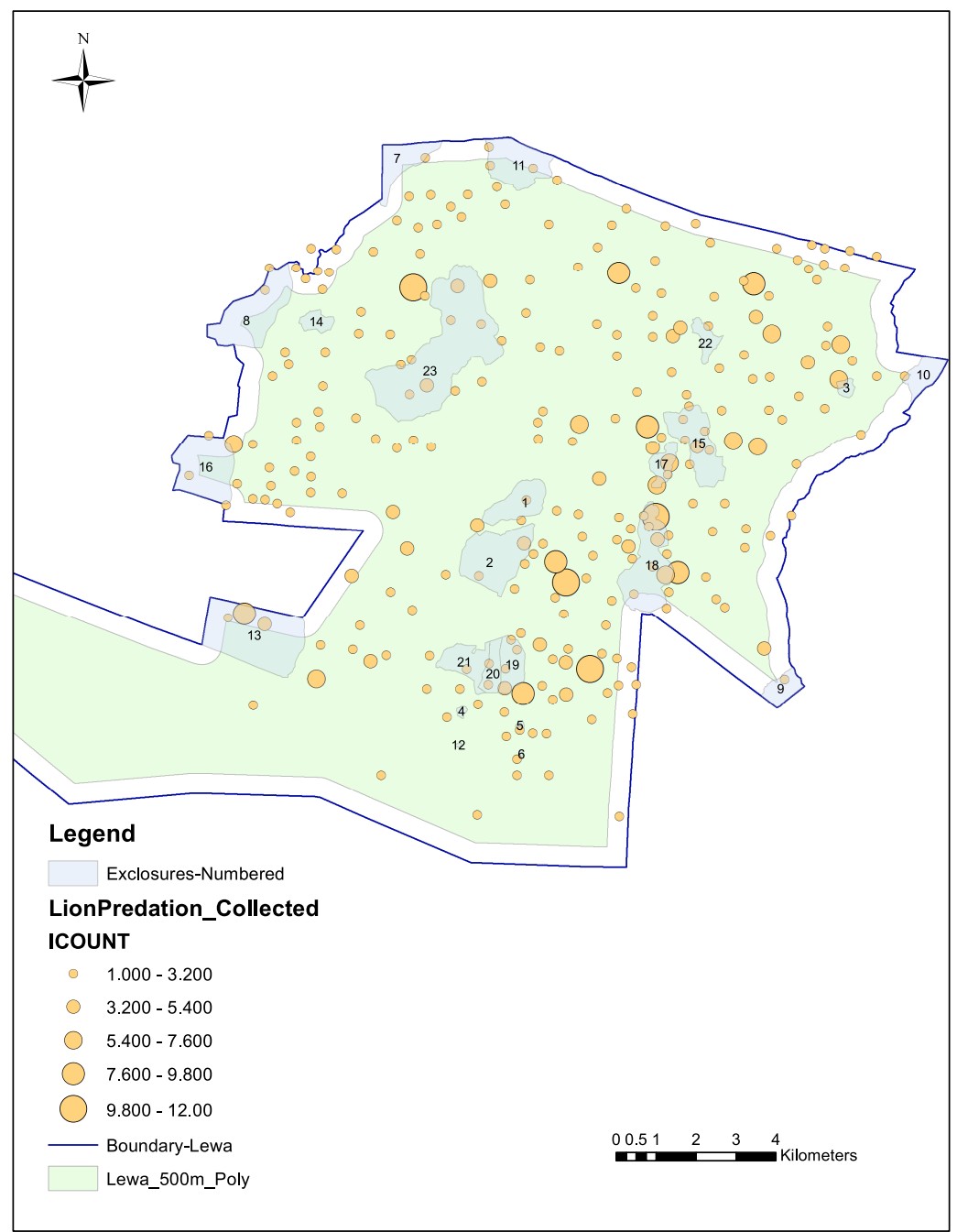

**Figure 2** Individual lion kills, collected at 100m tolerance, in relation to perimeter and exclosure fencing.

**Table 2** Actual versus expected kills near perimeter fencing and exclosures at the Lewa Wildlife Conservancy, 2005–2014.

| | Area (km²) | Lion kills | Kills per km² | Expected kills | Pearson's chi-squared | *p*-value (df 1)< |
|---|---|---|---|---|---|---|
| Total | 250 | 628 | 2.51 | | | |
| Within 100 m of perimeter | 9.52 | 29 | 3.04 | 23.91 | 1.08 | 0.2984 |
| Within 500 m of perimeter | 46.53 | 82 | 1.76 | 116.88 | 10.41 | 0.0013[*] |
| Exclusion zones | 27.47 | 121 | 4.40 | 69.00 | 39.18 | 0.0001[**] |
| +50 m buffer | 32.43 | 140 | 4.31 | 81.46 | 42.06 | 0.0001[**] |

**Notes.**
[*]Significantly smaller than expected based on area.
[**]Significantly greater than expected based on area.

In aggregate, the number of kills within the combined exclosure area and within the 50 m buffers is significantly larger than would otherwise be expected (see Table 2).

Although exclosures as a whole appear to attract more predation than would otherwise be expected based on the size of these zones, this phenomenon is not uniformly distributed across all exclosures. An individual assessment reveals that only a few exclosures seem to be significantly over-represented, i.e., showing a large differential of actual kills over the expected values: exclosures 15, 17 and 18 (see Table 3). Also of note, there were seven exclosures that had perimeter fencing as one border and none of these exclosures showed significantly over-represented lion predator kills.

## Prey selectivity index

We calculated a prey selectivity index (PSI) to compare the proportion of prey species inside the exclosures to that outside the exclosures (see Table 4). The PSI inside the exclosures generally mirrored the PSI outside the zones although we noted some increases in the preferences for waterbuck and warthog and decreases in the proportions of plains zebra and eland (both open plains species). Unsurprisingly, the preference for giraffe was lower (due to this species generally being excluded from the exclosures unless elephant have managed to break the electrical wire). Notably, the proportion of Grevy's zebra killed remained the same inside or outside the exclosures.

## Vegetation

Vegetation cover varied through the study site and was classified in 5 broad classes. We partitioned the vegetation cover and the number of lion kills in each cover class (see Table 5 and Fig. 3). Out of the 628 lion kills, 619 could be associated with a distinctive class of vegetation. Kills were under-represented in both extreme types of cover, i.e., open grasslands and in forested areas. Kills were over-represented in mixed habitat, where tree cover ranged from 2% to 40%, i.e., in woody grassland, shrubland and in woodland.

## Hot spot analysis

We aggregated the individual lion kill sites from 628 individual locations to 263 aggregated kill sites using the 100 m tolerances yielding a range from 1 to 12 kills at each aggregated

**Table 3  Individual exclosures actual number of kills versus expected based on area size.** Potentially meaningful differences in bold. (Lewa 2004–2014.)

| Zone ID | Area (ha) | Lion kills | Exp. kills | Actual-exp. | Area +50 m | Kills +50 m | Exp. kills +50 m | Actual-exp. +50 m | Min tree cover % |
|---|---|---|---|---|---|---|---|---|---|
| 6 | 1.2 | 0 | 0.03 | −0.03 | 4.3 | 0 | 0.11 | −0.11 | 21 |
| 12 | 1.3 | 0 | 0.03 | −0.03 | 4.5 | 0 | 0.11 | −0.11 | 40 |
| 5 | 3.2 | 1 | 0.08 | 0.92 | 7.6 | 1 | 0.19 | 0.81 | 21 |
| 4 | 5.4 | 0 | 0.14 | −0.14 | 10.9 | 0 | 0.28 | −0.28 | 21 |
| 3 | 15.9 | 0 | 0.40 | −0.40 | 24.6 | 6 | 0.62 | 5.38 | 11 |
| 14 | 32.8 | 0 | 0.82 | −0.82 | 46.1 | 0 | 1.16 | −1.16 | 21 |
| 22 | 37.7 | 3 | 0.95 | 2.05 | 55.3 | 3 | 1.39 | 1.61 | 12 |
| **17** | 48.6 | 15 | 1.22 | **13.78** | 66.5 | 15 | 1.67 | **13.33** | 11 |
| 20 | 49.1 | 4 | 1.23 | 2.77 | 67.2 | 4 | 1.69 | 2.31 | 21 |
| 9 | 55.4 | 1 | 1.39 | −0.39 | 71.9 | 1 | 1.81 | −0.81 | 21 |
| 10 | 61.6 | 1 | 1.55 | −0.55 | 80.3 | 1 | 2.02 | −1.02 | 6 |
| 19 | 72.5 | 8 | 1.82 | 6.18 | 92.0 | 8 | 2.31 | 5.69 | 21 |
| 21 | 86.1 | 3 | 2.16 | 0.84 | 113.2 | 3 | 2.84 | 0.16 | 21 |
| 7 | 96.8 | 1 | 2.43 | −1.43 | 125.0 | 1 | 3.14 | −2.14 | 11 |
| 1 | 103.8 | 1 | 2.61 | −1.61 | 127.2 | 1 | 3.19 | −2.19 | 21 |
| 11 | 150.7 | 5 | 3.79 | 1.21 | 177.5 | 5 | 4.46 | 0.54 | 11 |
| **15** | 158.2 | 12 | 3.97 | **8.03** | 190.5 | 15 | 4.79 | **10.21** | 21 |
| 2 | 204.3 | 6 | 5.13 | 0.87 | 235.9 | 6 | 5.92 | 0.08 | 21 |
| 16 | 207.0 | 1 | 5.20 | −4.20 | 236.9 | 5 | 5.95 | −0.95 | 1 |
| **18** | 216.7 | 29 | 5.44 | **23.56** | 256.0 | 30 | 6.43 | **23.57** | 21 |
| 8 | 220.6 | 2 | 5.54 | −3.54 | 256.8 | 4 | 6.45 | −2.45 | 16 |
| 13 | 342.1 | 14 | 8.59 | 5.41 | 380.7 | 14 | 9.56 | 4.44 | 21 |
| 23 | 576.2 | 14 | 14.47 | −0.47 | 643.1 | 15 | 16.15 | −1.15 | 18 |
| All | | 121 | 69 | | | 138 | 82 | | |

**Table 4  Prey Selectivity Index (PSI), inside versus outside exclosures.**

| Top prey species | Mean pop. from census 2004–2014 | Lion kills at study site | Lion kills outside exclosures (+50 m) | Lion kills inside exclosures (+50 m) | PSI outside exclosures | PSI inside exclosures |
|---|---|---|---|---|---|---|
| Plains z. | 1,069 | 253 | 204 | 49 | 0.26 | 0.13 |
| Grevy's z. | 380 | 127 | 99 | 28 | 0.37 | 0.37 |
| Giraffe | 207 | 60 | 51 | 9 | 0.31 | 0.06 |
| Eland | 181 | 43 | 35 | 8 | 0.18 | 0.07 |
| Buffalo | 378 | 29 | 25 | 4 | −0.37 | −0.60 |
| Waterbuck | 119 | 28 | 17 | 11 | 0.02 | 0.43 |
| Warthog | 132 | 25 | 12 | 13 | −0.21 | 0.46 |
| Impala | 881 | 23 | 15 | 8 | −0.83 | −0.69 |
| Oryx | 79 | 11 | 9 | 2 | −0.09 | −0.21 |

**Table 5** Vegetation cover and kill distribution using a supervised classification, (LANDSAT 8 data).

| Vegetation cover | Tree cover | Percentage of Lewa | Actual kills (619) | Percentage of kills | Expected kills | Chi-Squared ($df = 4$) | Prob< |
|---|---|---|---|---|---|---|---|
| Grassland | 0–2% | 25.7% | 123 | 19.7% | 159 | 8.13 | 0.087 |
| Woody grassland | 2–10% | 10.3% | 110 | 17.8% | 64 | 33.22 | 0.0001[**] |
| Shrubland | 11–20% | 33.7% | 268 | 43.3% | 209 | 16.84 | 0.002[**] |
| Woodland | 21–40% | 11.3% | 102 | 16.5% | 70 | 14.72 | 0.005[**] |
| Forest | 41%+ | 19.0% | 16 | 2.6% | 117 | 87.69 | 0.0001[*] |

**Notes.**

[*]Significantly LESS than expected.

[**]Significantly MORE than expected.

site. We calculated the Getis-Ord Gi⋆ statistic for each point of aggregated data using a search distance threshold of 5,796 m (as calculated by performing an incremental spatial autocorrelation procedure) and mapped these points in Fig. 4. Each predation location had its $Z$-score fall within one of seven bins, representing the statistical probability of significance, reported as a color ranging from blue (cold) to red (hot). Hot spots were mainly detected deeper inside the conservancy, although a few hot spots fell within the buffer areas near the eastern perimeter fence. Statistically significant hot spots were also detected within three of the exclosures (15, 17 and 18), some of these containing multiple hot spots. However, hot spots were also detected in many other areas throughout the conservancy. Of the 263 aggregated predation locations, 51 were classified as hot spots with a Gi⋆ score greater than 1.96 ($p$-value < 0.05) and of these 51 hot spots, 11 were located inside the exclosures (five in exclosure 15, and three in exclosures 17 and 18) and, none in exclosures with perimeter fencing.

## DISCUSSION

Predator–prey dynamics within fenced wildlife reserves can change in response to being held in a constrained habitat. Prey might avoid the fence, thereby creating an edge effect, while predators might drive prey into the direction of the fences, thereby reducing escape options. In this study we examined if predation cases were disproportionally found near perimeter fencing and found that, at the conservancy-level, predation events were not over-represented near the perimeter fence. On the contrary, our results indicated that there were fewer predation events than expected near the perimeter fencing, suggesting some edge effect. We suspect that human activity near the perimeter fence and the risk of human wildlife conflict discouraged herbivory near the boundary.

However, if edge effects were solely responsible for the predation pattern near the boundary fence then we would expect predation success to increase proportionally farther away from the perimeter fence. Our results showed that predation was proportionally lower within the 500 m buffer zone than it was within the 100 m buffer zone, suggesting greater hunting success nearest the perimeter fence. The results of the hot spot analysis also showed a few locations near the perimeter fence where predation was overrepresented. Taken together, these results suggest that predators might have higher hunting success in

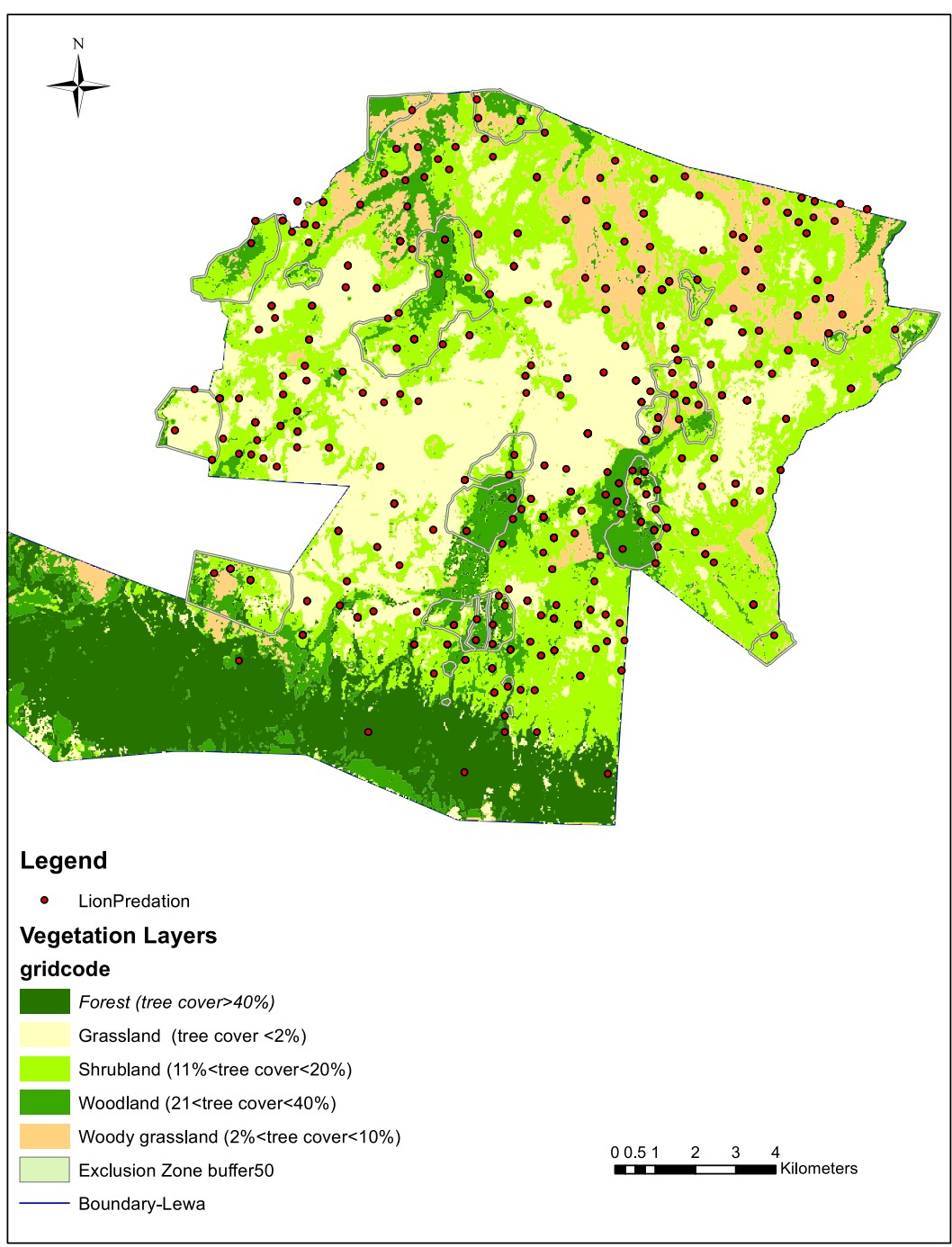

**Figure 3   Lion predation and vegetation cover.** Supervised classification of Landsat 8 imagery, 2014 data.

certain specific locations near the boundary fence although not at sufficient levels to impact the overall distribution of predation events. Further studies of hunting behaviour at these locations may shed light on the importance of the perimeter fence in hunting strategies of predators in fenced and semi-porous reserves.

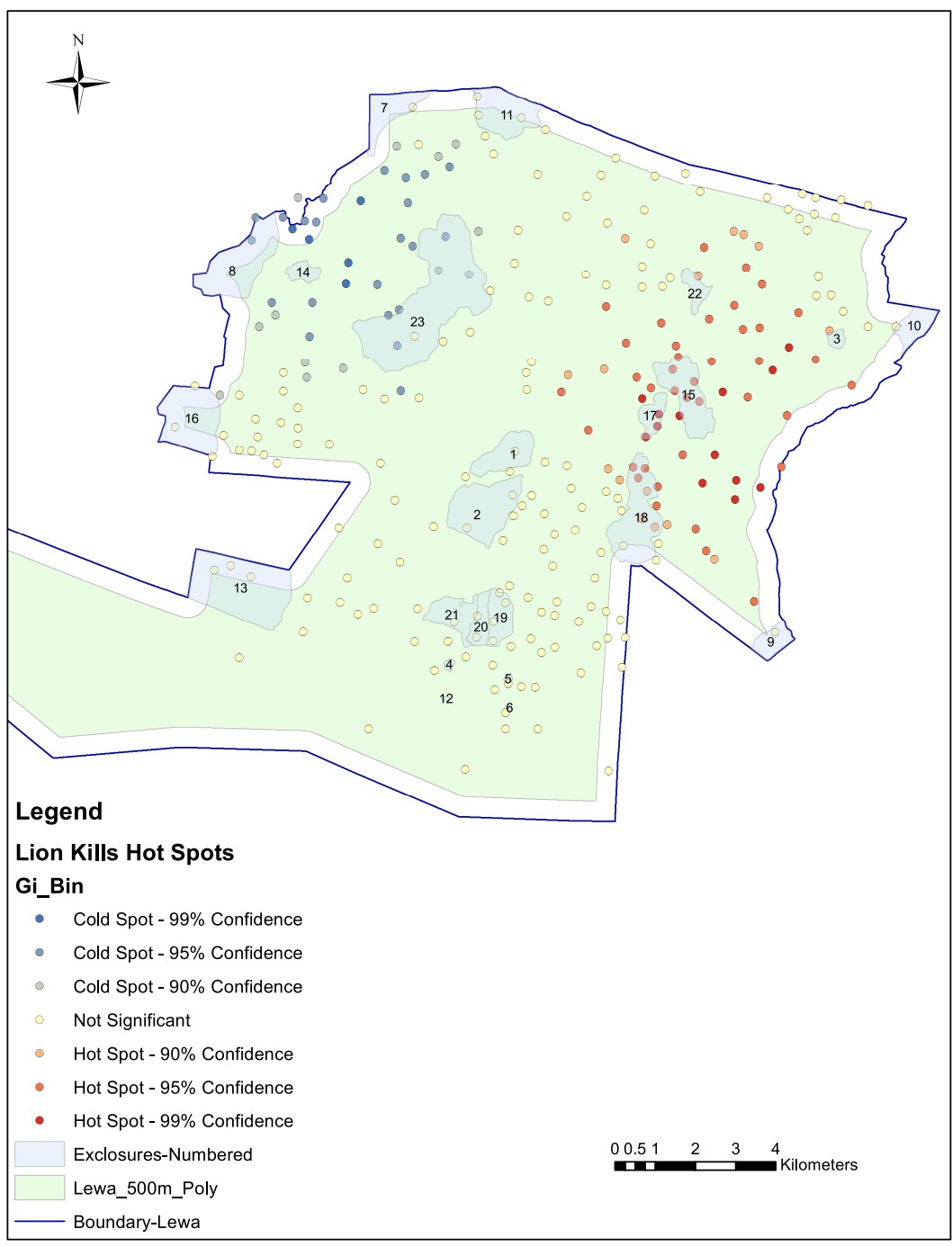

**Figure 4** **Hot spot analysis of lion predation using a zone of indifference conceptualization and a search distance of 5,796 m.**

With regards to elephant exclosures, we found that, at the conservancy level, predation events were over-represented in or near the exclosures. These results support the hypothesis that there was an increased risk of lion predation in and around exclosures compared to that in the rest of the conservancy. However, upon a more detailed hot spot analysis at the

exclosure level, we found that only a few of the exclosures contained hot spots while the majority of the hot spots were found outside the exclosures. Thus, the results of the hot spot analysis did not support the hypothesis that exclosures become prey-traps.

We also noted that predation was over-represented within certain vegetation cover categories consistent with other lion studies (*Funston et al., 1998*; *Hopcraft, Sinclair & Packer, 2005*) and these cover categories were found within the "hot" exclosures. These "hot" exclosures had been created around areas of somewhat already dense vegetation that has become progressively denser with time. However, most exclosures have not developed into fertile hunting grounds for lion possibly because foraging from other browsers has limited the vegetative growth (*Sankaran, Augustine & Ratnam, 2013*; *Staver & Bond, 2014*) or created thickets that are not conducive to lion hunting (*Tambling et al., 2013*).

Our findings should give managers the confidence to continue protecting some of the dense vegetation strands with exclosures knowing that there is no generalized increase in the risk of predation.

### Funding
The authors received no funding for this work.

### Competing Interests
The authors declare there are no competing interests.

### Author Contributions
- Marc Dupuis-Desormeaux conceived and designed the experiments, performed the experiments, analyzed the data, contributed reagents/materials/analysis tools, wrote the paper, prepared figures and/or tables, reviewed drafts of the paper.
- Zeke Davidson performed the experiments, contributed reagents/materials/analysis tools, wrote the paper, prepared figures and/or tables, reviewed drafts of the paper.
- Laura Pratt and Mary Mwololo performed the experiments, contributed reagents/materials/analysis tools, reviewed drafts of the paper.
- Suzanne E. MacDonald conceived and designed the experiments, performed the experiments, contributed reagents/materials/analysis tools, wrote the paper, prepared figures and/or tables, reviewed drafts of the paper.

### Field Study Permissions
The following information was supplied relating to field study approvals (i.e., approving body and any reference numbers):
All necessary permits were obtained for the described field study from the appropriate agencies (Kenya Wildlife Service Affiliation, KWS/BRM/5001, and Kenyan National Council for Science and Technology, NCST/RRA112/I/NIASI).

### Data Availability
Data was uploaded as Data S1.

## Supplemental Information

Supplemental information for this article can be found online at http://dx.doi.org/10.7717/peerj.1681#supplemental-information.

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
