# Peer review of "Testing the effects of perimeter fencing and elephant exclosures on lion predation patterns in a Kenyan wildlife conservancy"

_PeerJ, doi:10.7717/peerj.1681_

## Round 0.1 · original submission · Major Revisions

We have received two reviews for your manuscript and both reviewers found merits in your study but also raised important points that need to be revised. First and foremost, both reviewers suggested that predation events should be considered as independent and analysed accordingly (e.g. not as aggregated individual kill site). The apparent assumption that probability of carcass recovery is independent of habitat structure should also be justified and tested if possible. Moreover, it would be relevant to assess the possible connected effects of exclusion zone and boundary fence on your results. Finally, a few sections need to be clarified and, in particular, the introduction should be refocussed to better reflect the importance of the study and clarify the objectives.

·

Basic reporting

This paper reports long-term observations of predation within a fenced reserve, comparing the distribution of predator-killed animals with the location of elephant semi-proof fences that are used to protect trees and bushes from destruction. The sample size in terms of number of carcasses is impressive. My main concern is the apparent assumption that probability of carcass recovery is independent of habitat structure. It seems to me that the more forested the habitat, the less likely that a carcass may be found, especially in time to identify cause of death. Can this assumption be justified?

The Introduction should better present the relevance of this study: how widespread is the use of elephant exclosures in conservation areas, and why is there a concern about these areas serving as predator ambush sites?

Specific comments:

Abstract, line 34: but not at second glance? What does this mean: was it or was it not? It would be better to state that this was the result when all exclosure zones were pooled.

Lines 47-55: This is all true, but the paper is about exclosure zones, not perimeter fences, therefore the Introduction should be about those smaller exclosures.

Lines 86-87: OK, but at this point it is not too clear what was the first objective. That should be stated more explicitly earlier on in the Introduction.

Lines 129-130: not too clear what the relevance of this statement is, in relation to the objectives of the paper. Maybe the point here is that the exclosures were not entirely effective in allowing dense vegetation growth?

Lines 278-279. First, this sentence makes no sense – how can it be 0 of top 5 but 8 of top 10? Second, presumably this comparison is rather meaningless unless the relative area of exclosures and outside exclosure is accounted for.

Experimental design

Lines 151-153: not too clear why this was done - why not treat each predation as independent? I suspect this was necessary for the hotspot analysis, but then it should be better explained.

Line 165: presumably cheetah are not ambush predators? That should be made clear - in any case they accounted for fewer predation events.

Line 166: but would leopard kills not be more difficult to find, if they are hidden in trees?

Line 214: at this point, it remains unclear to me why we are presented with the ‘aggregated kill sites’.

Lines 240-242: written this way, it sounds like there were 2 selection criteria, and the second - "differences were large" - suggests a bias. I suspect that the only real criterion was the expected 5 kills or more, so this part needs to be better explained. Also, in the table it looks like the ‘real’ criterion was 4 kills or more (area 15).

Table 3: not clear why zones 19-21 were considered separately as they appear to form a single clump.

Fig. 4: there is a potential problem here, as many of the exclusion zones are adjacent to the boundary fence, where presumably access (or escape) was impossible - it would be useful to test if there were any effects of whether or not the exclusion zone involved a part of the boundary fence.

Validity of the findings

I am not sure that the Discussion makes sense. Lions are attracted to denser vegetation because predation there is easier. Exclosures lead to denser vegetation, and in some of them this paper shows that predation risk did indeed seem high. Therefore, some exclosures do create better hunting habitat for lions. Perhaps the point here is that good hunting areas in this reserve were not limited to the exclosure zones? This argument needs to be better articulated. Presumably, a conclusion from this research may be that if most of the undamaged vegetation is inside elephant exclosures, then those exclosures will have very high predator risk. Note also that unless this artificial habitat structure increases the overall level of predation inside the reserve, it is not clear that site-specific predation is a management concern.

·

Basic reporting

The article is well written and clear. There is an adequate introduction and background to show how the work fits into the field of knowledge. The paper is well structured and the figures are relevant. The raw data is available.

Experimental design

This is original primary research testing an interesting theory of the impact of fencing (and it also relates to corridors to an extent). My only concern about the methods is that the data are strongly biased by lion kills - and I suggest analysing the kills of each predator separately (where possible) to see if there are separate effects of each predator.

Validity of the findings

The data are good and well analysed (but see suggestion above). The conclusions make sense to me.

Additional comments

Aside from the reanalysis with each predator, I think the paper is very good.

Some minor suggestions:
- L47: I don't think this is restricted to 'private' conservancies??

- L55: see also Hayward and Kerley 'Fencing for Conservation ' in Biol Conserv.

I hope this is helpful.

Matt Hayward

---

## Round 0.2 · Minor Revisions

I agree with the reviewer that this is a much improved manuscript. I am thus willing to recommend its acceptance once you have dealt with the minor revisions suggested by the reviewer. In addition, you should indicate the names of statistical software/packages used for statistical analyses and also include the degrees of freedom for analyses presented in Table 5.

·

Basic reporting

This is a much-improved ms. i only have 2 minor comments.

Line 83 and elsewhere: technically, this means 'with an earlier date' and the correct word is 'preyed upon'. Maybe say 'predator kills' to avoid upsetting extreme anglophiles? Also, try to avoid the verb 'predated' elsewhere in the ms, for the same reason.

Line 143: I have no idea of how strong this voltage is - was it meant to give a slight shock to dissuade elephants and giraffe? It may be worth pointing that out. Did the perimeter fence have higher voltage?

Experimental design

Fine - nice sample

Validity of the findings

Looks good to me.

---

## Round 0.3 · accepted · Accept

All changes required were made and this version of the ms is now suitable for publication in PeerJ.